# Relationship between the Use of Social Networks and Mistrust of Mass Media among Mexican Youth: A Mixed-Methods and NLP Study

**Daniel Javier de la Garza Montemayor [1],\* and Xunaxhi Monserrat Pineda Rasgado [2]**

1   Departamento de Administración, Escuela de Negocios, Universidad de Monterrey, Av. Morones 4500 Prieto, San Pedro Garza García 66238, Nuevo León, Mexico
2   Centro Universitario de Ciencias Económico Administrativas, Universidad de Guadalajara, Periférico Norte N° 799 Núcleo Universitario, Los Belenes, Zapopan 45100, Jalisco, Mexico
\*   Correspondence: daniel.delagarza@udem.edu

**Abstract:** The relationship between trust and media consumption has been a subject continually studied in communication sciences. There are various studies that indicate that the greater the confidence in a certain medium, the greater the consumption. However, due to the consolidation of digital media and specifically social networks as the main sources of information among the young, the question of whether trust in these media affects their consumption arises again. To examine this relationship, a study was carried out with a mixed methodology. On the one hand, a survey was carried out in which young Mexican university students were asked about the media that they trusted the most and those that they used the most. On the other hand, interviews were conducted with university professors regarding their observations of the behavior of students in relation to usage and trust in these media. In general, the results allow us to conclude that trust in the media is not a variable that impacts the consultation of a specific media outlet given that, despite mistrust, there may be a high rate of usage.

**Keywords:** trust; social media; internet; youth; information

## 1. Introduction

The COVID-19 pandemic caused a transformation in practically all sectors of human life, and particularly made digital media essential for carrying out work, social, economic, and cultural activities. This makes the study of social networks increasingly relevant in the social sciences, and it is necessary to approach the subject matter from different perspectives (De la Garza and Peña-Ramos 2022). At first, these media seemed to bring with them a positive element towards democratization due to the use they were given in certain movements and social outbreaks such as the Arab Spring, 15M in Spain, or the #YoSoy132 movement (Candón 2013). Because of this, several investigations focused on the relationship between the use of electronic media and political participation.

However, today we have seen events in the world that have questioned whether the influence of digital media continues to positively strengthen democracy. Therefore, new questions have arisen along with different angles of research that need to be explored (Hall et al. 2018). The main objective of this investigation is to determine if pre-existing trust in media is a variable that determines the consumption of political content, whether traditional or digital. This approach is necessary as currently informational biases, as well as misinformation, have had unexpected impacts on social life.

## 2. Literature Review

### 2.1. Confidence and Consumption of Conventional and Unconventional Media

On the issue of trust, conclusions have been reached that trust in a medium is essential for an individual's decision to continue its usage (Gainous et al. 2019). If people trust a

medium, they tend to consume more information from it. This can have an implication in the vision that citizens have about democracy, and potentially in their decision to act on matters of a public nature.

This process, ironically, is stronger in places where democracy has limitations. In times of growing political polarization and the emergence of populist alternatives on both the left and right, research has shown that those citizens who support anti-establishment alternatives distrust traditional media more than they do virtual media (Liu et al. 2020). But among those citizens who maintain more moderate positions, they usually value journalistic aspects (in all types of media) such as political analysis and the questioning of the excesses of public power. But the information from previous studies can only apply to a specific place. Arguably, during the presidential election in Mexico in 2018, social media did not damage trust in institutions in a greater proportion than traditional media, as was expected (Echeverría and Mani 2020).

However, another study agrees that the credibility that people grant to a certain medium corresponds to the journalistic credibility it has (such as the presence of sources) (Llamero et al. 2019). In a context of wide polarization, users may be motivated to participate (or comment) in media that have a different ideological position than their own, because in this way, they consider that they can balance the information.

There are similarities with research carried out during the confinement in Spain during COVID-19, where it was found that the information that citizens consume more, and also consider more reliable, come from those media that express similar ideologies to their own (López et al. 2020). There may come a situation in which citizens' distrust of the information they receive becomes unanimous. This is a scenario in which people declare that they distrust all sources of information that are presented, representing a deep crisis of institutional trust, which ends up transcending ideologies and political parties across the spectrum (Gualda and Rúas 2019; Marques et al. 2022).

The trust that users have in the same social media is a factor that influences the loyalty that they may have to a particular site. This factor is so important that some argue that trust plays an important mediating role between user loyalty and the satisfaction they get from the site (Sadiq et al. 2020). Without the presence of trust, there is no relationship between the other two factors.

*2.2. Media and Perceptions*

Media generate perceptions that influence how people think. Even when we perceive a transition in which mass media are progressively losing their monopoly on influencing public opinion, a study shows us that they still maintain an important influence (Jacuński et al. 2019). On the other hand, social media have had an enormous responsibility building the perceptions that users have of mass media. An investigation concluded that, during the presidential elections in the United States in 2016, social networks contributed to the questioning of the credibility of traditional media (Weeks et al. 2019).

But these conclusions differ from another study that argues that political polarization is the consequence of audiences that consider that the information they receive from the media with which they maintain an ideological identification is the most reliable. To the extent that a medium demonstrates a different narrative, audiences tend to distrust the source (Kelly 2019; Cetina and Martínez 2019).

This situation is valid for both traditional and digital media. While in the past it was an editorial stance that can reinforce the convictions of viewers, today, platforms such as YouTube work in a similar way. The algorithm that suggests content to users usually recommends videos that are in the same argumentative line as those that have been previously selected. The suggested alternatives strengthen the preconceived perceptions of citizens who use this social network (Lin et al. 2016; Vihalemm et al. 2019).

There is a study that reinforces this position. It maintains that there are some audiences that interpret the information in the following way: If what a medium describes is aligned with their convictions, they choose to consider that it behaves with professionalism and

objectivity. Otherwise, they might choose to have a critical attitude towards the information presented (Javaid and Elahi 2014).

It is also possible to argue that the transition from communication media to digital media entails a significant risk because the younger generations, who largely consume digital media, are not used to checking the news they receive (Catalina-García et al. 2019). This could represent a serious problem, since the perceptions obtained about what is happening around them come from sources that, in many cases, are weak (Sup and Kaye 2019).

### 2.3. Social Networks as an Informative Medium

As part of the transition of mass media that has occurred in recent years, it has been noted that social networks are a means of communication that inform citizens about relevant information. According to a study, there is a positive relationship between the usage of social networks and knowledge about political issues (David et al. 2019). Following the same logic, the conclusions of another study are similar, as it shows that users who consult social media for news have a greater knowledge of issues of public interest (Kim and Dennis 2019; Mohamed et al. 2020).

The researchers found a clear differentiation between those who use traditional media for public affairs and those who consult social networks, in which the second group was more politicized than the first. It is also important to recognize that social media can help some users persuade others about a specific agenda. This influence is more noticeable when the users receive information from friends and family (Dewi and Satyawan 2022).

One of the main differences between digital and traditional media has to do with the plurality of voices that are recognizable in the dissemination and positioning of specific information. In the view of some, the traditional media were more efficient in setting the public agenda, while in the time of social networks, more actors are involved in the interpretation and spreading of information (Yerlikaya 2020; Arianto et al. 2019).

As social media have become a relevant and, in many cases, even a primary source, awareness of how they can contribute to misinformation has increased. Acknowledging a new reality also implies noticing some of the main risks presented by digital media (Tucker et al. 2017). These include practices related to information verification methods (Fink and Gilich 2020; Domínguez 2015). Therefore, in current times, the change in the consumption of information on politics is not questioned, but there are doubts about the challenges that arise with this new reality. Fake news and misinformation constitute a serious challenge for democracy (Moreno and Ziritt 2019).

### 2.4. Young People and Consumption of Social Networks

The importance of social networks has been progressively recognized as a tool used by young people to organize themselves and discuss public affairs (Rubio-Romero and Espinosa 2015; Domínguez 2015). These media have enabled interaction that can be both collective and individual. These media have also changed civic involvement, giving rise to movements that have a different dynamic from those of the past (Alonso López and Bolinches 2020).

Due to its popular use among youth, WhatsApp has been found to be one of the applications most valued by young people because it allows them to make contact in a timely manner (Huang et al. 2021). There are also studies that indicate that use has been given to Instagram as a platform capable of contributing to the expansion of knowledge of students by allowing them to continue to learn about subjects that interest them in a didactic manner (Fuster-Gullén et al. 2020). This is mainly because young people value interacting in a very visual way (Bustamante Pavez 2017).

There are similarities with an experiment in which it was confirmed that social networks can contribute to improving student learning by connecting them with hypertext links (Gavilán et al. 2017). In this way, students can encounter different sources of information, which is something that favors the fact that they can delve into the topics they investigate. From a commercial point of view, brands that seek to connect with young

people through digital media have had to employ a different language from the one used in a traditional marketing strategy. A study pointed out how young people tend to pay more attention to those brands that behave like another user than to a company that only seeks to promote its products (Cano et al. 2017).

Digital media have also functioned as a platform for activism and to defend social causes. However, the results of one study show that these cases are generally a minority (Hoffmann and Lutz 2021). Most of the Spanish students that were surveyed remain skeptical of actively participating in social media or expressing themselves on these issues. These results are largely consistent with other research in which young people remain skeptical of participating in politics, even when they have extensive digital resources at their disposal (Serrano and Serrano 2014; Valerio and Serna 2018).

However, it is also important to note that social media can also have a negative effect on the young. In another survey carried out, it was reported that a significant proportion of young people surveyed had suffered situations of harassment through social networks (De la Garza et al. 2019). Social media can also end up generating a sense of disconnection among youth (Díaz et al. 2013).

## 3. Materials and Methods

The main objective of this paper is to analyze the relationship between the variables of consumption and trust in media based on a data analysis of a polling instrument applied to young university students between the ages of 18 and 25 during the 2018 electoral period in Mexico. Likewise, it seeks to describe the patterns of consumption and trust, specifically observing the relationship between the level of consumption of digital media and the degree of trust or distrust in them. For this, a hypothesis test was performed using the Mann–Whitney U statistic.

In the same way, we applied text mining to analyze a group of interviews carried out with professors where they were asked for their observations regarding the same phenomenon, consumption, and trust in the media of their students. The analysis was carried out by applying computational artificial intelligence (AI) techniques, known as Natural Language Processing (NLP), with the aim of obtaining new empirical conclusions through mathematical models of Machine Learning, and thus assess and propose the use of these new tools for the analysis of qualitative data in social research in Spanish. For both approaches, the Jupyter Notebook computing platform was used, and the programming language used was Python in its version 3.9.7. Based on previous research, it is advisable to have more than a single point of view regarding perceptions on the use of technology. Therefore, it can be argued that it is useful to have both students and professors' testimony, because students are the actors of this transition, but professors have been a witness of this process.

On the one hand, students provide valuable information about their own behaviors and perceptions of the media, which allows for a deeper understanding of the degree of trust that influences their consumption of political information. On the other hand, teachers can provide a more objective and observational perspective of their students. Since teachers interact with students in an educational setting, they can provide insight into how students seek and consume political information in a more formal context. In addition, teachers can gain a broader view of trends in media consumption among young people, which can help contextualize the results of the student survey.

### 3.1. Quantitative Approach

The data used for this first part were obtained through a survey of university students in which they were presented with a list of media, both traditional and digital, and using a Lickert-type scale, they were asked to indicate the respective level of usage, as well as in which they place greater trust or distrust. The design of the questionnaire took into consideration previous studies on the subject, and it had already been validated in other contexts (Krishnan and Rogers 2014). The sample was non-probabilistic, for convenience,

in which 804 young university students residing in the state of Nuevo León, the Valley of Mexico and Oaxaca participated.

The survey was applied through the Google Forms platform, so, the responses were obtained electronically. Likewise, these were applied between 30 March and 27 June of the year 2018, since, according to Mexican electoral legislation, this is the period in which political parties are allowed to carry out electoral campaign acts. For the pre-processing and analysis of the data, the Pandas, Seaborn and Matplotlib libraries were used. For the hypothesis test, mainly the stats module of the Scipy library was used. Media consumption variables were treated as ordinal variables since a Likert-type scale was used for their measurement. On the other hand, the trust variables were treated as nominal, since the possible answers to these measures were: (1) I trust it, (2) I do not trust it and (3) I don't know.

Hypothesis

To carry out the hypothesis test concerning the relationship between the variables of consumption and trust in the media, it was first necessary to confirm the type of distribution of the data. For this, the Kolmogórov–Smirnov test as shown in Table 1 was carried out, which establishes the following:

**H$_0$.** *The sample follows a normal distribution;*

**H$_1$.** *The sample follows its own distribution.*

**Table 1.** Results of the Kolmogorov–Smirnov test.

| | |
|---|---|
| Statistic | 0.916 |
| *p*-value | 0.00 |

According to the results of the normality test, the result of the statistic is equal to 0.916 and the corresponding *p*-value is 0.0. Since the *p*-value is less than 0.05, it is possible to reject the null hypothesis; that is, we have enough evidence to say that the sample data do not come from a normal distribution.

Once confirmed that our data follow a non-normal distribution, it was possible to formulate the hypothesis that allowed us to understand the type of relationship between the variables of consumption and trust in the media. The statistic used was non-parametric, in this case, the Mann–Whitney U test. The approach included the following:

**H$_0$.** *The degree of confidence is not related to the level of media consumption; that is, the variables are independent;*

**H$_1$.** *The degree of confidence is related to the level of media consumption; that is, the variables have a certain measure of dependency.*

*3.2. Qualitative Approach*

The data used for this second part of the analysis are textual and come from a series of interviews with 16 research professors: 5 from the state of Nuevo León; 6 from the Valley of Mexico and 5 from the state of Oaxaca, all with more than five years of experience in the classroom and with a field of study in the social sciences. The professors were selected based on the following criteria: they must be Social Sciences professors with at least five years of experience. We wanted to make a match on the perceptions of both actors. The interviews were of a structured type so that they had the same common thread according to the analysis categories established in the survey (TensorFlow n.d.).

The questions were aimed at gathering information that teachers could have observed and perceived in the behavior of their students regarding the two dimensions considered in this study: (1) The level of consumption of traditional and digital media by university

students and (2) The degree of confidence they appear to have regarding both categories of media.

Data analysis and preprocessing were performed with the help of Pandas, Re, NLTK and Gensim libraries. The implementation of these tools is part of the study approach called Natural Language Processing (NLP), a field that combines linguistics and artificial intelligence (AI) so that computers can understand human or natural language (Elekes et al. 2020).

The NLP techniques applied to this study comprised 4 main phases: The first consisted of a stage of cleaning and purifying the text of the transcribed interviews. At this point, capital letters, tildes and special characters were removed. In this way, the algorithm could label each of the terms more precisely. In the second phase, a tokenization process was carried out, which consisted of classifying and separating strings of characters into entities called tokens.

For this work, two tokenization processes were considered: the first in sentences and the second in words. At this point, stop words were also eliminated. These are words that do not add value to data labeling, such as prepositions or articles, and therefore do not capture the essence of the words and phrases that help natural language processing. In the third stage, the Wordcloud library was used to create a graph that could proportionally visualize the frequencies of the most recurring words in the text, also known as a word cloud.

Finally, the fourth phase consisted of the embedding of text or Embedding. This is a deep learning technique or Deep Learning that consists of a dense and continuous representation of words in a low-dimensional vector space. The advantage of this vector representation is the possible encoding of general semantic and syntactic relationships between words, assigning similar words to nearby points in the representation space (Elekes et al. 2020).

For this, the Gensim library was imported, and the Word2vec pre-trained neural network was used. This tool converts the words into vectors, and once the algorithm has been trained with our data, it could represent the words according to their context. The export of our model and resulting files were loaded in the Jupyter Notebook and code web application. We use the open embedding projector at http://projector.tensorflow.org (accessed on 26 August 2022). With this tool, it was possible to show the visualization of the embedding work easily and automatically and thus observe the closeness of words according to the cosine similarity.

## 4. Results

### 4.1. Quantitative Results

According to the results of the survey (Table 2), students state that they have greater distrust in the media that are presented digitally or electronically. The data showed that almost 60% of students distrust the information they receive through emails, followed by social networks (57.6%) and blogs (57.1%). Likewise, it was observed that the media that enjoy greater trust are those considered traditional media. The results show that the media that are most trusted among university students are: the printed magazine (61.1%), the written press (55.5%) and the radio (49.3%). However, this generalization does not extend to television. In this sense, it was observed that, on the contrary, television is the only traditional means of communication to which young people show greater distrust.

**Table 2.** Confidence in the media of university students from Nuevo León, Valle de México and Oaxaca.

|  | **Majority Option** | **Full Percentage** |
|---|---|---|
| Trust in TV (News) | I do not trust it | 48.63 |
| Trust in TV (Programs) | I do not trust it | 50.12 |
| Trust in Radio (News) | I trust it | 49.25 |
| Trust in Radio (Programs) | I trust it | 40.42 |
| Trust in e-mails | I do not trust it | 59.95 |
| Trust in web pages | I do not trust it | 44.28 |
| Trust in blogs | I do not trust it | 57.09 |
| Trust in Social Networks | I do not trust it | 40.67 |
| Trust in Social Media | I do not trust it | 57.59 |
| Trust in Press | I trust it | 55.47 |
| Trust in Magazines | I trust it | 61.07 |

N = 804. (1 means "He trusts", 2 means "He does not trust, and 3 means "He does not know").

Regarding consumption, young university students state that the media they consume use the most are social networks, as shown in Table 3, (M = 3.23, STD = 0.93), with 51.37% responding to consult them "Quite a lot" (the highest value they could choose), followed by the digital press (M = 2.44, STD = 1.07) with 32% of young people who say they consult these media "A lot". Regarding the least consumed media, the results were: television (M = 1.32, STD = 0.96), with 43.91% of young people stating that they consume it "Little", followed by the written press (M = 1.34, STD = 0.9), with 41% indicating the same response, in addition to radio (M = 1.31, STD = 1.0), with 39.4% of young people also stating that they consume it "Little". In general, young people consume traditional media to a lesser extent, having a greater preference for digital and electronic sources.

**Table 3.** Media consumption of university students from Nuevo León, Valle de México and Oaxaca.

|  | **TV** | **Written Press** | **Digital Press** | **Radio** | **Printed Magazine** | **Social Media** | **Blogs** |
|---|---|---|---|---|---|---|---|
| Never | 19% | 18% | 4% | 21% | 21% | 0% | 9% |
| Little | 44% | 41% | 14% | 39% | 37% | 5% | 22% |
| Somewhat | 26% | 33% | 32% | 30% | 28% | 17% | 36% |
| A lot | 9% | 7% | 33% | 8% | 12% | 26% | 21% |
| Quite a lot | 3% | 2% | 17% | 2% | 2% | 51% | 12% |

N = 852. The table shows the percentage of each degree of consumption according to the medium. The minimum value is 0 and the maximum value is 4. There are 5 values: Never (0), Little (1), Somewhat (2), A lot (3) and Quite a lot (4).

Hypothesis Test

Table 4 shows the results concerning the previously stated hypothesis test, which seeks to understand the relationship between the consumption and trust variables. Because the p value is greater than 0.05, it is possible to accept the null hypothesis, which states that the degree of confidence is not related to the level of media consumption; that is, the variables are independent.

**Table 4.** U test results.

|  |  |
|---|---|
| Statistic | 44,163 |
| *p*-value | 0.199 |

*4.2. Results of Natural Language Processing*

4.2.1. Word Frequencies

As can be seen in Figures 1 and 2, teachers generally observed that the trust and consumption of university students reside mainly in digital media and specifically in

social networks. Throughout the interviews, there was a clear consensus among those interviewed that the main sources of information for young university students are social networks. Furthermore, contrary to what the young people say, their teachers perceive that their trust also lies in said media, since some of them mentioned certain episodes in which their students were victims of misinformation.

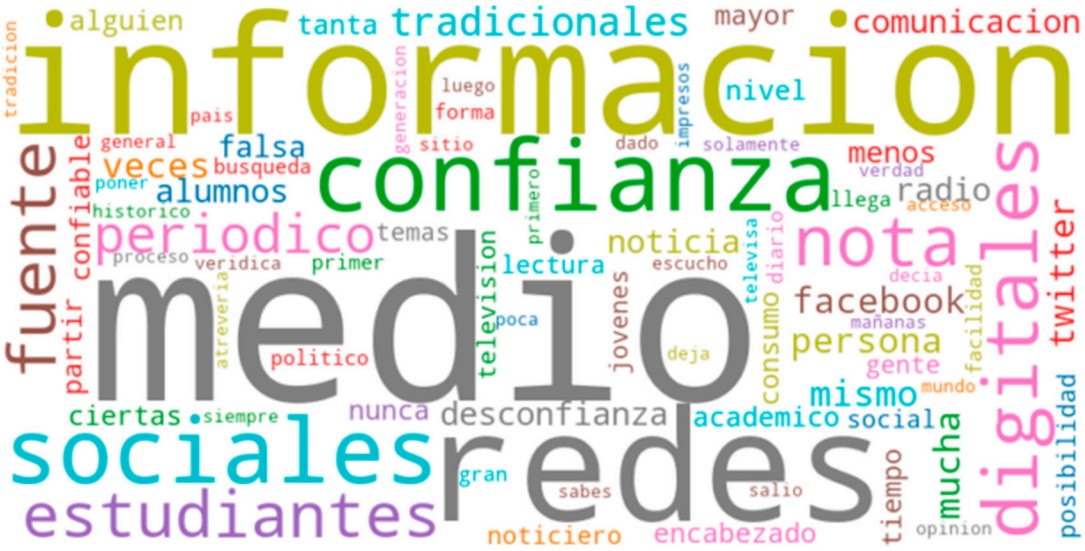

**Figure 1.** Frequency of words about trust in the media.

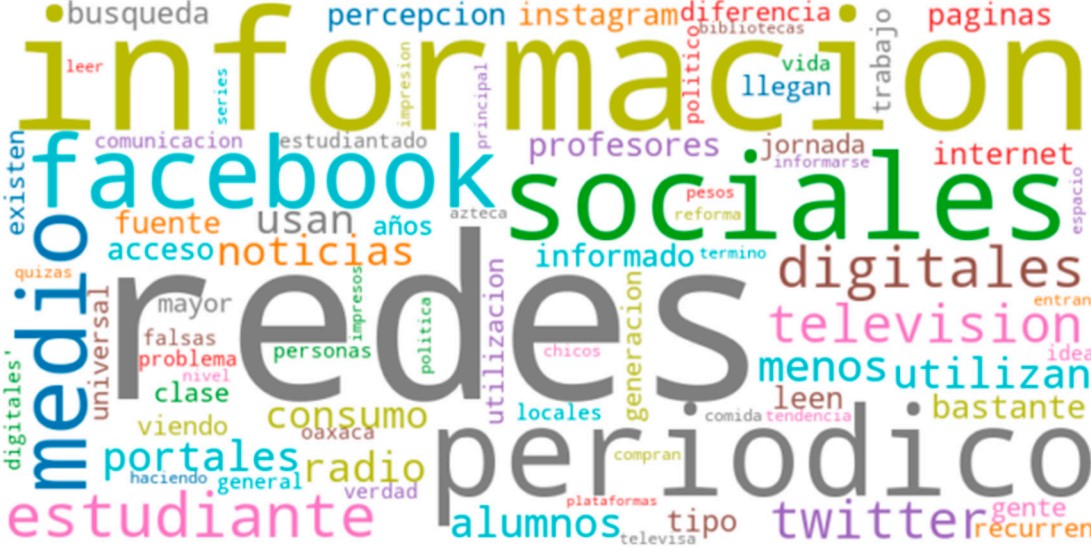

**Figure 2.** Frequency of words about media consumption.

In order to better analyze the results obtained with the embedding process, it was decided to visualize a couple of specific vectors. In other words, from the selection of two specific words, "consumption" and "trust", it was possible to visualize the words that our algorithm was able to relate. Figure 3 shows the vector space and the words close to the word "consumption".

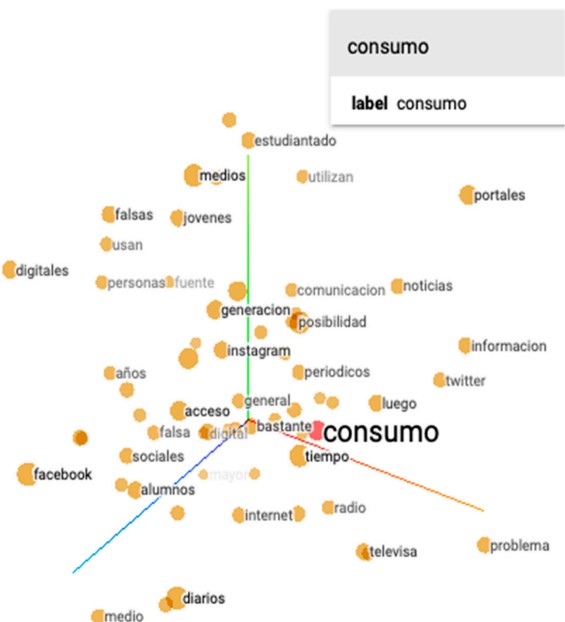

**Figure 3.** Visualization of the "consumption" vector.

4.2.2. Embedding Visualization

In order to better analyze the results obtained with the embedding process, it was decided to visualize a couple of specific vectors. In other words, from the selection of two specific words, "consumption" and "trust", it was possible to visualize the words that our algorithm was able to relate. Figure 3 shows the vector space, and the words close to the word "consumption". Specifically, the closest words are the following: "access", "media", "facebook", "generation", "portals", "news", "students", "time", "prints" and "information".

Likewise, Figure 4 shows the vector space where the word "trust" is highlighted. Specifically, the closest words were: "people", "themes", "media", "portals", "generation", "digital", "headline", "social", "print" and "consumption".

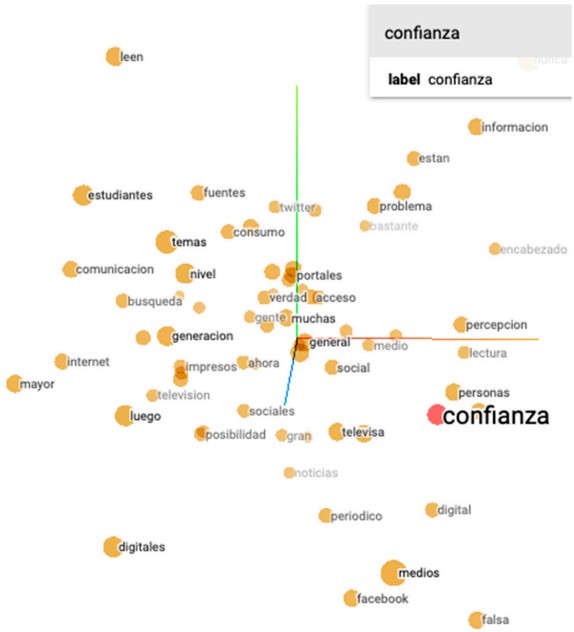

**Figure 4.** Visualization of the "trust" vector.

## 5. Discussion and Conclusions

The results of the survey allowed us to anticipate the possible relationship that exists between trust and media consumption. It was observed that, in the case of television, usage was minimal and at the same time a degree of distrust was evident. It was also observed that the radio or the press had a higher degree of trust; however, usage was not higher but, on the contrary, was lower. Likewise, the phenomenon also presented changes in reference to social networks, since despite the clear distrust, these are the media most consumed by Mexican youth.

The results of the U test allowed us to provide more statistical evidence to support the hypothesis that in these times of the digital boom, young natives of the digital world do not consider trust as a factor that determines their consumption decisions. With this evidence regarding the independence between these two variables, the results of the qualitative approach seem to offer a possible assumption that would be worth exploring in future research, which is that the consumption of these digital media, and especially social networks, is not precisely determined by trust, but rather by an aspect of easy access to information that identifies this generation.

In this sense, obtaining the information that young people obtain through digital means seems to be due to a question of optimizing the time and effort involved in searching for and comparing information. Therefore, it is interesting that the embedding model that was trained for this research has related words such as "access", "time" or "header". The teachers stated that they observed that the students had much information at hand on social networks; however, beyond inquiring about the events related in headings, the headings of the notes seemed to be enough to satisfy students' information needs: a fact that may represent a problem affecting young people's ability to distinguish false information and consequently their ability to make informed decisions in the political arena. Finally, it is hoped that the research presented here will serve to motivate the use of these types of computational tools within social research and thus enrich the methods of qualitative data analysis.

The findings confirm the two hypotheses raised for the development of this research. H1 confirms a trend that was visible in past research, among which are those that affirm that there is a change in the consumption of information on politics (Moreno and Ziritt 2019). At the same time, this research coincides with the research that found the positive relationship that exists between consumption of social networks and the politicization of young people (David et al. 2019). As the qualitative results indicate, young people learn about political and social events, even if they initially access platforms for entertainment purposes.

In addition, it is also possible to affirm that the results prove H2, since even though the young people surveyed expressed greater distrust towards digital media, it is precisely these media that are consumed more frequently. Based on the data obtained, it was not possible to confirm other trust findings, since the highest levels of trust that young people expressed, in traditional media, did not correspond to their consumption (Gainous et al. 2019). However, these findings could partially confirm that, despite the progressive decrease in the presence of the mass media, they still maintain an important influence on public opinion due to their migration to social networks (Jacuński et al. 2019). Of course, based on the data collected, the latter does not apply to television, since this was the only medium in which there is a negative correlation in both trust and consumption.

Another important conclusion of this study is that when young people say that they have distrust in social networks, they possibly refer to the electronic medium as such and not to the network of acquaintances with whom they interact in that space and from whom they obtain and relay information. Social networks have already been confirmed as a means of information and communication; however, the dynamics of their operation are very different from the traditional model of the mass media.

As some point out, in the logic of the traditional media, there was only one channel that issued the message that would be received by a wide audience. In this sense, the establishment of a specific editorial position was much easier to identify (Yerlikaya 2020).

However, in the dynamics of social networks, there is no clear editorial line, since within the same medium, multiple broadcast channels of content can be presented; therefore, the determining factor turns out to be the interpretation of the information by each user.

Consequently, this phenomenon causes the parameters of trust to be transformed; that is, it causes users to go from trusting an established medium, to trusting a network of contacts and followers who, in turn, also play a role as a channel for generating and distributing information.

Having said this, it would be possible to understand why young people argue that they do not trust social networks, even though qualitative data shows the opposite. In other words, it is possible that when young people say they have distrust towards social networks, they do so by understanding them from a generalized conception of a digital tool. But in the use and consumption of these, the trust they show is not towards the medium, but towards their own network of contacts.

According to this view, users may consider members of their network as reliable sources. However, there is also a risk that a member of your network could distribute false information. As some research reveals, users of social networks are increasingly aware of this risk, so it is possible that the knowledge of the existence of this phenomenon is what makes them mistrust social networks in general. Obviously, this argument gives rise to new studies that would allow us to confirm this approach with greater certainty. In general, it seeks to contribute to and motivate the study of the need to better understand how young people consume political information, since this has a direct impact on democratic systems that function as an important foundation for the making of informed political decisions (Tucker et al. 2017).

**Author Contributions:** Conceptualization, D.J.d.l.G.M.; methodology, D.J.d.l.G.M.; software, X.M.P.R.; validation, D.J.d.l.G.M. and X.M.P.R.; formal analysis, X.M.P.R.; investigation, D.J.d.l.G.M. and X.M.P.R.; resources, D.J.d.l.G.M. and X.M.P.R.; data curation, X.M.P.R.; writing—original draft preparation, D.J.d.l.G.M. and X.M.P.R.; writing—review and editing, D.J.d.l.G.M. and X.M.P.R.; visualization, D.J.d.l.G.M. and X.M.P.R.; supervision, D.J.d.l.G.M. and X.M.P.R.; project administration, D.J.d.l.G.M. All authors have read and agreed to the published version of the manuscript.

**Funding:** This research received no external funding.

**Institutional Review Board Statement:** This study used informed consent for all participants (students and teachers).

**Informed Consent Statement:** Informed consent was obtained from all subjects involved in the study.

**Data Availability Statement:** Data on mixed methods is available upon request.

**Conflicts of Interest:** The authors declare no conflict of interest.

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
