# Peer review of "Relationship between the Use of Social Networks and Mistrust of Mass Media among Mexican Youth: A Mixed-Methods and NLP Study"

_socsci, doi:10.3390/socsci12030179_

Round 1

Reviewer 1 Report

The paper shows a good grasp of the relevant literature. The references are strong.

The methods section should provide more information. For example, was the survey created by the authors? What questions were included? Why were these questions included? Was the questionnaire validated? How? With what results?

The qualitative analysis seems innovative and original, but it appears disconnected from the quantitative analysis. You have two different populations with two different questionnaires. You will need to create a convincing argument to explain how the two sections of the study are connected. More information is needed regarding the participants. How were the university professors recruited? Why were they selected? A deeper analysis would have provided more compelling information than the word clouds.

Overall, I believe the qualitative section of the study is more original. Consider focusing only on these data, omitting the quantitative portion of the study, explaining in greater detail the use of NLP for data analysis, and doing a more in-depth analysis of the interviews.

Some proofreading is necessary. See, for example, lines 48, 58-59, 102 and the last line in Table 2.

Author Response

Dear reviewer,

Thank you very much for your observations. They have been all addressed in a new version of the article. We will answer all the questions that have been raised.

First, the survey was made by one of the authors, based on previous studies. That has been clarified in a revised edition. The questionnaire was validated in previous studies one of the leading authors has made before, in other countries.

The professors were selected with the following criteria: Social Sciences professors with at least five years of experience. The argument for having both students and professors’ testimony if because students are the actors of this transition, but professors have been a witness of this process. We wanted to make a match on the perceptions of both actors. In México, university students (both from public and private universities) have been involved in social movements that have been inspired by social networks. Professor have witnessed.

We are grateful for the suggestion on concentrating only on the qualitative study. We made a survey for students, and interviews with professors. The latter was decided because there are considerably less professors that students, but also, we wanted to know their perceptions about the change observed in the last five years. We feel it is necessary to keep data from both studies.

Thank you for the proofreading suggestions. We revised that information that was pointed out and overall.

Reviewer 2 Report

I suggest carrying out a more complete analysis taking into account the effect of other variables. Also the opinion of teachers about young trust and consumption of media does not seem to be relevant for and empirical purpose.

Author Response

Dear reviewer,

Thank you very much for your observations. They have been all addressed in a new version of the article. The analysis that you have suggested has been improved.

We made a survey for students, and interviews with professors. The latter was decided because there are considerably less professors that students, but also, we wanted to know their perceptions about the change observed in the last five years. We feel it is necessary to keep data from both studies. The empirical results of the interviews are valid and reveal a low level of sophisticated political knowledge, despite the prior distrust in social media. Students may not consciously believe in the content of social media, but they end up believing in it because it is their main source of information. 

We hope to have clarified these points and that our response is helpful to you. We appreciate your time and consideration, and we are available for any further questions or doubts. 

Best regards,

Authors

Reviewer 3 Report

Attached intended for both authors and editors.

Author Response

Dear reviewer,

We hope this letter finds you well. First and foremost, we would like to thank you for taking the time to review our article and for providing us with your observations and suggestions. 

We would like to take a moment to respond to your comments about establishing the causality principle between media consumption and trust. Our research actually aims to precisely point out the lack of causality between these variables, which differs from previous studies conducted on these topics, which only focused on traditional media. In this research, we consider social media as a means of obtaining political information. 

Regarding social media and digital media, they not only do not fit into this dimension, but they also represent a new area of investigation. We suggest that the preferred consumption of these media is not determined by trust, but by factors such as easy access. This finding opens the door to future research seeking to understand what determines the consumption of digital media for obtaining political information. 

We also would like to mention your comment about the lack of consideration of the variables of polarization and political knowledge. We acknowledge that they are important and relevant topics, but the data used in our research did not focus on them. Nevertheless, although trust does not determine the use of social media, it remains the main source of information for university students, which may represent a risk in the level of political polarization and political knowledge. 

Finally, we would like to highlight that the empirical results of the interviews are valid and reveal a low level of sophisticated political knowledge, despite the prior distrust in social media. Students may not consciously believe in the content of social media, but they end up believing in it because it is their main source of information.  We hope to have clarified these points and that our response is helpful to you. We appreciate your time and consideration, and we are available for any further questions or doubts. 

Best regards,

Authors

Round 2

Reviewer 1 Report

I am satisfied with the additional information regarding the method used. 

I believe you can do a better job justifying the inclusion of two dissimilar populations with two different instruments. Your argument is okay, but it can be written in a more forceful manner.

On your Likert scale, your options are Gives me confidence/ Does not give me confidence. I think you are translating directly from Spanish: Me da confianza, no me da confianza. However, in English, the idea of "gives me confidence" is not the same. A better translation would be I trust it (them) I don't trust it (them).

Notice that media is a plural word and needs a plural verb. Media are... Media have....

Author Response

Dear reviewer, we are thankful for your observations. First of all, we added a more substantial justification for the inclusion of two different populations (lines 178-185). We feel that the arguments have been significantly improved, thanks in part to your comment. We also corrected the translation on the Likert scale you pointed out. Once again thank you, because your comments have definitely improved our text.

Reviewer 2 Report

None

Author Response

We are thankful for the observations. We took the opportunity in improving our text and it is definitely better after we took into consideration your comments.

Reviewer 3 Report

The revisions make the paper better, especially as it relates to the discussion and conclusions. I still have three concerns. First, I feel the literature review could do a better job of laying out the theoretical issues and making it clear where this study fits. If I understand the overall thrust, trust was once linked to media use, but that relationship no longer holds. Second, while the concluding discussion is good, I would like to see more about the implications of the findings, particularly as it relates to politicization, polarization and democracy. The issue of democracy is raised in the beginning, while the authors' response notes the likely impact of the findings on politicization and polarization. Finally, I still do not see much that relates to the 2018 presidential election. While the study was conducted during the period, the survey does not incorporate anything about the election. Consequently, including this in the title may be a bit misleading.    

Author Response

After giving much thought, we decided to change our title so it would not be as misleading. It no longer references to the election in the title. We think the new one is more suitable for the article. We also added some comments on the discussion and conclusions (lines 404 and 435) to deal with politization (the ability to make an informed decision), and hoy consumption of political information has an impact on democratic systems. We appreciate your feedback, since we believe it has improved our text.